# In Silico Analysis of MiRNA Regulatory Networks to Identify Potential Biomarkers for the Clinical Course of Viral Infections

**DOI:** 10.3390/ijms262010100

**Published:** 2025-10-16

**Authors:** Elena V. Mikheeva, Kseniya S. Aulova, Georgy A. Nevinsky, Anna M. Timofeeva

**Affiliations:** SB RAS Institute of Chemical Biology and Fundamental Medicine, Novosibirsk 630090, Russiaamaya.rain.nsu@gmail.com (K.S.A.);

**Keywords:** miRNA, regulatory networks, biomarkers, viral infection, SARS-CoV-2, HIV-1, Flaviviridae, long COVID, miRWalk, Cytoscape, gene network visualization

## Abstract

MiRNA expression profiles exhibit notable alterations in numerous diseases, particularly viral infections. Consequently, miRNAs may be regarded as both therapeutic targets and markers for the development of complications. MiRNAs can significantly influence the modulation of immune responses, offering an extra layer of regulation during viral infections. In this study, miRNAs associated with viral infections were analyzed using an in silico approach. Computer modeling predicted a number of miRNAs capable of influencing the functionality of specific components of the immune system. As a result, 242 miRNAs common to the three types of infections were identified. A network of miRNA-gene regulatory interactions, encompassing 502 nodes (224 miRNAs and 278 genes) and 2236 interactions, was developed. Within this network, subnetworks were identified that are involved in the operation of specific connections in the immune response to viruses. For each step of the immune response, the miRNAs involved in governing these processes were examined. These predicted miRNAs are of particular interest for further analysis aimed at establishing the relationship between their differential expression and disease symptom severity. The obtained data lay the foundation for identifying the most promising molecules as predictive biomarkers and the subsequent development of a diagnostic system.

## 1. Introduction

RNA viruses are characterized by high evolutionary rates due to their frequent replication cycles and low fidelity of RNA polymerase, resulting in mutations that may be linked to high pathogenicity in humans. Thus, it is not unexpected that RNA viruses are responsible for the most recent significant epidemics. SARS-CoV, SARS-CoV-2, and MERS-CoV are highly pathogenic coronaviruses that induce severe acute respiratory illness [1,2]. Presently, HIV-1 continues to be a considerable worldwide public health issue. According to UNAIDS, approximately 88.4 million individuals have been infected with HIV, and approximately 42.3 million have died from AIDS-related illnesses since the beginning of the epidemic (https://www.un.org/en/global-issues/aids accessed on 22 August 2025). Over the past few years, there have been significant threats to global health because of consistent outbreaks of infectious diseases from the *Flaviviridae* family, including Dengue virus (DENV), Zika virus (ZIKV), West Nile virus (WNV), and tick-borne encephalitis virus (TBEV). The DENV is transmitted to humans by mosquitoes, causing roughly 400 million infections and tens of thousands of deaths annually [3,4]. Over the last decade, ZIKV has infected millions, causing substantial epidemics in the tropics and extending to other regions [5,6].

The advent of the COVID-19 pandemic, an unanticipated event, severely tested the medical field and exposed multiple systemic weaknesses in the treatment and prevention of infectious diseases and their secondary effects. It is plausible that humanity may confront a new viral threat in the future, which could trigger another epidemic. Therefore, a search for predictive markers regarding the clinical course of viral infections is highly relevant. MiRNAs can be considered as representative markers. These elements control numerous biological mechanisms, thus contributing substantially to various processes [7]. Notable changes in the miRNA expression profile are observed in diverse diseases, encompassing viral infections [8,9,10,11,12,13,14]. Certain miRNAs can be assessed as important regulators of immune responses, inflammatory mediators, and cytokines [15,16]. Consequently, miRNAs can be considered as therapeutic targets in viral infection, as well as markers for the progression of complications. Identifying shared miRNA regulatory pathways across various infections is crucial not only for understanding the infections themselves but also for addressing novel diseases that may emerge in the future.

The objective of this work was to identify miRNAs with the potential to predict the severity of viral infections. The miRWalk database was used to identify miRNAs associated with viral infections of various etiologies, specifically flavivirus, coronavirus, and HIV. A total of 242 miRNAs were found to have altered profiles in the three viral infections investigated. The correlation between the identified miRNAs and genes was analyzed. Genes associated with the functioning of the immune system during viral infection were identified by gene ontology analysis. A miRNA-gene regulatory interaction network, encompassing 502 nodes (224 miRNAs and 278 genes), with 2236 interactions, has been established. Subnetworks connected to the function of single links in the immune response to viral infection were detected in this network. The involvement of miRNAs in regulating these processes has been assessed for each facet of the immune response. The research has successfully identified miRNAs that impact the functions of individual immune system components and can be employed to predict the severity of clinical progression in viral infections, including potential infections that humanity may face in the future.

## 2. Results

### 2.1. miRNAs Associated with Viral Infections

The miRWalk database was used to explore miRNAs linked to viral infections of different origins [17,18,19]. Here, the focus is on flavivirus, coronavirus, and HIV infections. The miRNAs were filtered according to the following selection criteria: (1) a predicted probability of binding to the target greater than 0.95, (2) binding of the miRNAs to the 3′-UTR, and (3) presence in at least one database: targetScan [20], miRDB [21], and miRTarBase [22]. In animal models, a fraction of the miRNA sequence is known to bind to the 3′-untranslated region (3′-UTR) of the target mRNA, which results in diminished gene expression [23,24]. Consequently, our focus was directed towards predicted miRNAs that interact with this region. Given the availability of over 10 miRNA target prediction programs that utilize different computational methods [25], miRNAs present in a minimum of one of the other three databases were selected.

The miRWalk database was used to extract the data on miRNAs associated with flavivirus infections caused by DENV, TBEV, WNV, and ZIKV. Of the four members of the Flaviviridae family, DENV has been the subject of the most extensive research. Consequently, it is unsurprising that the greatest number of miRNAs has been documented for this virus. Figure 1A illustrates the quantity of filtered miRNAs (miRWalk data) across the four flaviviruses. Eight miRNAs were found to be common to DENV and TBEV (miR-12118, miR-22-5p, miR-4437), DENV and WNV (miR-4443, miR-671-5p, miR-8085), DENV and ZIKV (miR-12131, miR-1273h-5p) infections. Our hypothesis is that the infrequent overlap in miRNAs across flavivirus infections stems from the relative lack of miRNA research in these viral infections. The high prevalence of miRNAs associated with Dengue fever detection results from intensive investigations into DENV infection, a prominent flavivirus. As an illustration, a search in PubMed (https://pubmed.ncbi.nlm.nih.gov/, accessed on 17 April 2025) revealed 20,357 publications for the query “dengue fever,” whereas the query “tick-borne encephalitis” yielded a result of only 2073 publications. Therefore, the absence of miRNA crossovers in the four flavivirus infections is explained by the lack of investigation into the miRNAs that regulate various genes in infections caused by viruses within the *Flaviviridae* family. For subsequent analysis, all miRNAs were combined and designated as “miRNAs associated with flavivirus infections.”

A similar approach was used to study miRNAs related to coronavirus infections, employing three search terms in the miRWalk database: “Coronavirus infectious disease,” “Middle East respiratory syndrome (MERS),” and “Severe acute respiratory syndrome.”

The search results for “Coronavirus infectious disease” and “Severe acute respiratory syndrome” were consolidated and termed SARS. Only eleven miRNAs have been associated with MERS, a likely consequence of the limited number of existing studies. The number of miRNAs associated with SARS was found to be significantly higher, accounting for 685. Only one miRNA (miR-3173-5p) was identified for both diseases (Figure 1B). For subsequent analysis, all miRNAs were aggregated and designated as “miRNAs associated with coronavirus infections.”

A search of the miRWalk database with the phrase “human immunodeficiency virus infection disease” was conducted to analyze miRNAs relevant to HIV infection, which resulted in the identification of 1152 miRNAs.

Consequently, three lists of miRNAs were generated.

### 2.2. Characterization of miRNAs in the Context of Different Viral Infections

Viral infections are known to be associated with changes in the expression of various miRNAs, with several mechanisms of host antiviral response by miRNAs having been identified [26] (Figure 2). Viruses are also known to encode miRNAs, but a discussion of viral miRNAs is beyond the scope of this study.

(1)MiRNAs can inhibit viral replication by directly binding to the viral RNA genome. For example, 873 miRNAs that can target the SARS-CoV-2 genome have been detected by in silico methods [27].(2)MiRNAs can participate in regulating infection pathways. For example, it has been shown that a number of miRNAs can regulate the expression of ACE2, a cell membrane-associated protein that is required for SARS-CoV-2 virus entry into the host cell. miR-125b directly targets the 3′-UTR of ACE2 mRNA, thereby regulating its expression [28].(3)MiRNAs can play a role in regulating the innate and adaptive immune systems.

The first two mechanisms on the list are virus-specific. A third mechanism indicates that several of these regulatory pathways may be common among viral infections. For example, large amounts of miRNAs can regulate the expression of IL-1β, IL-6, and IL-8 because there are numerous miRNA binding sites in the 3′-UTR of IL-1β, IL-6, and IL-8 mRNAs [29]. MiRNAs may also modulate the expression of cytokine signaling pathways and inflammatory responses during viral infection [30]. It is thus of interest to investigate miRNAs involved in regulating the immune response cascade in diverse viral infections.

Next, we compared the three miRNA lists to identify common miRNAs. The study identified 242 miRNAs among the three infectious disease types (Figure 3). A total of 877 common miRNAs were identified for coronavirus infection and HIV infection. Regarding flavivirus infectious diseases, a significantly smaller number of miRNAs were found in common with coronavirus infectious diseases (254 miRNAs) and HIV-infection disease (295 miRNAs). The dataset used for further analysis comprised 242 miRNAs.

Next, using miRTarBase [31] and TargetScan [32,33] databases, target genes for 242 miRNAs common to different viruses were identified, and their gene ontology (GO) was analyzed. These miRNAs can influence viral infection through one of the three pathways listed in Figure 2. The correlation between the viral genome (Figure 2A) and the control of infection pathways (Figure 2B) may vary substantially depending on the specific virus. Thus, given the similarities in immune responses to viral infections, the focus was directed to miRNAs involved in regulating various components of the human immune system.

### 2.3. Analysis of miRNA-Gene Interactions and Gene Ontology

Analysis of miRNA-gene interactions was performed using Cytoscape 3.10.2. It has a modular structure, and applications (plugins) allow for additional functions to enhance analysis. By employing the CyTargetLinker 4.0.0+ plugin [34,35] for analyzing 242 miRNAs, a miRNA-gene regulatory interaction network was constructed, comprising 14,507 nodes (242 miRNAs and 14,265 genes) and 124,211 interactions.

Subsequently, a Gene Ontology (GO) analysis was performed. Selection was made of genes associated with diverse elements of the immune response to viral infection. Consequently, a network comprising 502 nodes (224 miRNAs and 278 genes) and 2236 interactions was generated. Next, separate subnetworks were identified, enabling the characterization of miRNAs influencing the regulation of distinct components of the immune response to viral infection.

#### 2.3.1. MiRNAs Associated with the B-Cell Response

B-cell development involves several stages: from the early stages of B-cell differentiation in the bone marrow in the absence of antigen to later stages after interaction with antigen in the periphery [36]. Differential miRNA expression of B cell development indicates a regulatory role for these miRNAs at each developmental stage. A number of studies have demonstrated the involvement of miRNAs in early B cell development coordination [37,38,39]. This is succeeded by the release of B lymphocytes to the periphery and their interaction with antigens. Upon exposure to an antigen, some activated naïve B cells differentiate directly into short-lived cells that primarily produce IgM antibodies. Other B cells enter the follicle of secondary lymphoid organs such as lymph nodes and spleen to establish a germinal center and eventually differentiate into high-affinity IgG-producing plasma cells and memory cells [40]. Therefore, subsequent stages of differentiation are crucial for characterizing the function of miRNAs in regulating immune processes during viral infections.

GO analysis allowed 29 miRNAs to be identified that are involved in the regulation of B cell activation related to immune response (GO: 2312) (Figure 4). Included among these miRNAs are miR-29a-3p, miR-29b-3p, miR-486-3p, and others. In vitro deletion of miR-29a-3p has been shown to correlate with reduced B cell proliferation and antibody production [41].

The generation of antibodies is strongly associated with the advanced stages of B-cell development. Antibodies circulating in the blood and lymph neutralize pathogens, enhance phagocytosis, and participate in the destruction of cells affected by viruses. The study examined gene ontologies related to the production (GO: 2381) and regulation (GO: 2889) of the immunoglobulin-mediated immune response, as well as the immune response mediated by circulating immunoglobulin (GO: 2455). A gene ontology analysis identified 42 genes regulated by 144 miRNAs, including miR-146a-5p, miR-181a-5p, miR-486-3p, miR-650, miR-4689, miR-3175, miR-766-3p, miR-29a-3p, and others (Appendix A). Earlier findings established the expression of miR-181a-3p to be elevated simultaneously with B-cell activation [42,43].

#### 2.3.2. MiRNAs Associated with T-Cell Response

The immune response mediated by T cells is central to protecting the host from invading pathogens, such as viruses. Pathogen-infected cells can be directly eliminated by cytotoxic CD8+ T lymphocytes, which are the predominant type of T cell [44]. CD4+ T lymphocytes orchestrate the immune response by activating diverse immune cells, comprising B lymphocytes, CD8+ T lymphocytes, and phagocytic antigen-presenting cells [45]. Treg cells, specifically CD4+CD25+FoxP3+ cells, also can suppress unwanted immune activation and prevent detrimental immune-mediated inflammation and tissue injury [46].

Twelve genes govern T cell activation associated with the immune response, and three genes are involved in T cell proliferation. These genes are regulated miRNAs, including miR-146a-5p, miR-486-3p, miR-25-3p, miR-29a-3p, miR-29b-3p, miR-325-3p, miR-9-5p, miR-92b-3p, miR-98-5p (Figure 5).

MiR-29b-3p, miR-25-3p, miR-29a-3p, miR-32-5p, miR-363-3p, miR-367-3p, miR-92a-3p, and miR-92b-3p impact T cell function by controlling the expression of the EOMES transcription factor, which is essential for the differentiation of cytotoxic CD8+ T cells during immune responses by upregulating lytic effector genes [47,48].

Native antigens undergo fragmentation by antigen-presenting cells, followed by display of the fragments on the cell surface via major histocompatibility complex (MHC) molecules, enabling T-cell receptor recognition [49]. Infected cells present intracellular antigens, including viral ones, as part of the MHC class I complex. The regulation of MHC antigen processing and presentation during viral infection involves four genes: *AP3B1*, *B2M*, *CD1A*, and *CD1D*. The genes are, in turn, regulated by the following 15 miRNAs: miR-1-3p, miR-186-5p, miR-22-5p, miR-302b-5p, miR-302d-5p, miR-325-3p, miR-335-5p, miR-340-5p, miR-3664-5p, miR-3929, miR-4419b, miR-4478, miR-4522, miR-6739-3p, and miR-9-5p (Figure 6). Elevated miR-1-3p levels were correlated with the severity of COVID-19, with the highest levels corresponding to extended hospitalizations and reduced survival [50].

#### 2.3.3. MiRNAs Associated with the Natural Killer Cell Response

Natural killer (NK) cells are members of innate immunity that provide, among other things, defense against viruses by secreting a number of cytokines and killing virus-infected cells by initiating their apoptosis [51]. Over 400 and 300 miRNAs have previously been identified in human and murine NK cells, respectively, with the expression of a number of miRNAs found to alter upon NK activation [52,53]. The findings of this study indicate that, during viral infection, NK (GO: 2323) cell activation is governed by four genes, the regulation of which is influenced by sixteen miRNAs, including let-7b-5p, miR-25-3p, miR-29a-3p, miR-29b-3p, miR-363-3p, miR-367-3p miR-92a-3p, and others (Figure 7).

miR-29b has previously been shown to exhibit elevated expression in NK cells and to be critical for both the transformation of NK precursors into immature NK cells and the terminal maturation and function of NK cells [54,55,56]. Therefore, miRNAs could potentially regulate innate immunity by modulating natural killer cells.

#### 2.3.4. MiRNAs Regulating the Cytokine Response

MiRNAs have been computationally analyzed [57] and shown to predominantly target transcription factors, cofactors, and chromatin modifiers. In contrast, molecules within the innate immune pathway, such as Toll-like receptors (TLRs), chemokines, and cytokines, are generally not targeted by miRNAs. MiRNA binding sites were predicted in nine of the interleukin (*IL1-29*) genes under analysis. Examination of twenty interleukin receptor genes revealed high-probability miRNA target sites in just two. However, another study revealed that clusters of adenine- and uridine-rich elements (AREs) were identified in the 3′ untranslated regions (UTRs) of mRNA molecules encoding cytokines [58]. AREs recruit various ARE-binding proteins (ARE-BPs) that modulate mRNA stability and/or translation [59]. Despite the absence of miRNA target sites in the 3′-UTRs of most cytokines, miRNA can regulate cytokine expression by targeting ARE binding proteins, thus influencing the half-life of numerous cytokines [58]. Moreover, miRNAs can also modulate the expression of a range of cytokines through regulating transcription factors and elements of signaling pathways, which results in cytokine mRNA transcription. The current research established a comprehensive network of miRNA-gene interactions involved in modulating cytokine responses during diverse viral infections. We have identified 48 genes and 395 interactions related to the cytokine-mediated signaling pathway (GO: 19221). Due to the visualization and interpretation challenges presented by the resulting network, three subnetworks were identified: (1) regulation cytokine gene, (2) regulation cytokine receptor gene, and (3) cytokine signaling pathways regulation.

##### Regulation of Cytokine Gene

A regulatory network was constructed to show the regulation of cytokine gene during viral infection (Figure 8). Twelve cytokine genes are subject to modulation by twenty-six miRNAs, including let-7 family miRNAs, as well as miR-1-3p, miR-122-5p, miR-125a-5p, miR-146a-5p, miR-16-5p, miR-181a-5p, miR-186-5p, miR-335-5p, miR-340-5p, miR-34a-5p, miR-3662, miR-374a-5p, miR-374b-5p, miR-4279, miR-4458, miR-4500, miR-9-5p, and miR-98-5p.

IL-6 is implicated in the mechanisms of both acute and chronic inflammation, such as those observed in autoimmune conditions [60]. This cytokine promotes plasma cell survival and immunoglobulin secretion through the JAK/STAT3 signaling pathway, also promoting the differentiation of CD4+ cells into Th17 lymphocytes while inhibiting differentiation into Treg [61,62,63]. The regulation of IL-6 involves let-7a-5p, let-7c-5p, let-7f-5p, miR-146a-5p, miR-98-5p, miR-335-5p, miR-9-5p, and miR-1-3p.

The data presented here indicate that the *IL-10* gene is regulated by the let-7 family miRNAs, as well as miR-34-5p, miR-374b-5p, and miR-374a-5p. The cytokine in question suppresses myeloid cells, thereby modulating inflammation to minimize tissue damage, which is also significant for immune defense against viral infections [64]. At the same time, IL-10 stimulates proliferation, B lymphocyte differentiation, and antibody secretion and participates in the pathogenesis of autoimmune diseases such as systemic lupus erythematosus [65].

##### Regulation of Cytokine Receptor Gene

A regulatory network was constructed to illustrate the regulation of cytokine receptor gene in the context of viral infection (Figure 9). 16 cytokine receptor genes are modulated by 69 miRNAs, including the let-7 family, and also miR-1228-3p, miR-125a-5p, miR-146a-5p, miR-16-5p, miR-29a-3p, miR-29b-3p, miR-325-3p, miR-335-5p, miR-34a-5p, miR-34b-3p, miR-377-3p, miR-485-5p, miR-508-5p, miR-515-5p, miR-766-3p, miR-9-5p, miR-92a-3p, miR-98-5p, and others. Therefore, the function of several cytokines (IL-1, IL-6, IL-10, IL-17, and others) is also influenced by miRNAs through regulating receptor gene for these signaling molecules.

##### Regulation of Cytokine Signaling Pathways

The products of several genes are involved in the direct function and regulation of cytokine signaling pathways, including the JAK, STAT, SOCS, TRAF, IRAK, and other protein families. MiRNAs are involved in the regulation of these genes, which in turn leads to the regulation of the cytokine response. A miRNA network was constructed for a number of major genes involved in regulating cytokine signaling pathways (Figure 10).

The JAK/STAT signaling pathway can activate over 50 cytokines and growth factors. When activated, JAKs are responsible for cytokine receptor phosphorylation and then recruit STAT proteins [66]. The JAK2-dependent signaling pathway regulates crucial pro-inflammatory cytokines such as IL-6, IFN-γ, and GM-CSF [67]. JAK2 is regulated by three miRNAs: miR-125a-5p, miR-16-5p, and miR-335-5p.

STAT proteins are implicated in regulating transcription and mediate cellular responses to signals from cytokine and growth factor receptors [68,69,70,71]. STAT3, for example, functions as a regulator of the inflammatory response by modulating the differentiation of naive CD4+ T cells into Th17 or Treg T-helper cells [72,73]. The regulation of the *STAT3* gene during viral infection involves twenty-four miRNAs, including let-7 family miRNAs, miR-181a-5p, miR-29a-3p, miR-29b-3p, miR-98-5p, miR-125a-5p, miR-296-5p, miR-340-5p, miR-370-3p, miR-4458, miR-4500, miR-4516, miR-665, miR-6807-5p, miR-6845-3p, miR-6893-3p, miR-92a-3p and others. The *STAT1* gene is regulated by three miRNAs: miR-145-5p, miR-146a-5p, and miR-34a-5p.

Upon activation, STATs dimerize and enter the nucleus, prompting the expression of suppressor of cytokine signaling (SOCS), and SOCS proteins bind to phosphorylated JAK and its receptor, thereby negatively regulating the JAK-STAT signaling pathway [66]. SOCS1 acts as a crucial negative regulator of IFN-I, IFN-II, and other cytokine signaling pathways [74,75]. SOCS1 downregulates cytokine signaling through the inhibition of the JAK/STAT signaling pathway. During viral infection, the *SOCS1* gene is regulated by sixteen miRNAs, including the let-7 family, miR-98-5p, miR-122-5p, miR-222-3p, miR-4266, miR-4458, miR-4500, miR-4695-5p, miR-4779, and others.

The current findings are consistent with prior research, which indicated that miRNA let-7 and miR-98 binding sites were detected within the 3′UTR regions of *SOCS1*, *STAT3*, *IL-6*, *IL-10*, and *IL-13* transcripts [57].

TRAF molecules are involved in various signaling cascades, functioning as central regulators of immunity and inflammation. TRAFs exhibit the ability to interact with the cytoplasmic domains of cytokine receptors, such as IL-1β, IL-2, IL-6, IL-17, IL-18, IL-33, IFN type I, IFN type III, GM-CSF, M-CSF, and TGF-β, and also with TLRs, NLRs, RIG-I-like receptors, and C-type lectin receptors [76]. TRAF2 is a prerequisite for TNFα to activate MAPK8/JNK and NF-κB. Mice with a global TRAF2 deficiency (TRAF2 −/−) experience early lethality as a result of an inflammatory phenotype, which is marked by increased serum TNFα levels [77]. TRAF2 is regulated by the following six miRNAs: miR-34a-5p, miR-4279, miR-4459, miR-4695-5p, miR-4722-5p, and miR-6081. TRAF6 modulates cellular proliferation, differentiation, and apoptosis via the activation of NF-κB, JNK/p38, PI3K/AKT, and AP-1 pathways, as well as regulating innate and adaptive immunity, oxidative stress, and inflammation [78,79]. TRAF6 is regulated by 24 miRNAs, including miR-146a-5p. miR-146a-5p can suppress IL-6 and IL-1β secretion by modulating the IRAK1/TRAF6 pathway [80].

IRAK kinases serve as downstream signal transducers subsequent to receptor activation, exemplified by TLR pathogen recognition receptors or IL-1 receptors (IL-1R) [81]. Mutations in the IRAK1 protein have been detected in patients who are highly susceptible to severe infections [82]. The *IRAK1* gene is modulated by three miRNAs: miR-146a-5p, miR-222-3p, miR-92a-3p, while *IRAK2* is modulated by twenty-eight miRNAs, including miR-146a-5p, miR-16-5p, miR-34b-3p, miR-485-5p, miR-508-5p, miR-665, miR-766-3p, miR-873-3p, and other miRNAs.

Given the above, it can be assumed that a number of miRNAs may be involved in regulation of both the genes of cytokines themselves and the genes of participants of signaling pathways that are triggered by these cytokines. Alterations in cytokine profiles, including those stemming from diverse infections, may result in various complications, such as cytokine storms, and the initiation of autoimmune processes [60,83,84].

## 3. Discussion

Successful binding of a miRNA to its target typically results in translational repression and/or degradation of the mRNA [85]. Human miRNA is thought to be capable of regulating several hundred different mRNAs. Alternatively, a single mRNA can be the target of several miRNAs simultaneously [20]. MiRNAs offer a method of modulating rather than completely suppressing protein levels within the cell. The reduction in transcript levels resulting from the targeting of miRNAs to a specific mRNA typically does not exceed threefold, and this modest variation seems to correlate with translation levels [86]. The association between miRNAs and various diseases is being actively investigated because they are currently considered promising diagnostic markers and therapies.

MiRNAs offer multiple advantages when employed as biomarkers. First, miRNAs are stable molecules that are resistant to degradation [87]. The stability of miRNAs makes them appropriate for routine clinical analysis. Second, miRNAs can be detected in various biological fluids, including blood and saliva [88]. Third, miRNA quantification is feasible through techniques commonly employed in clinical laboratories, including quantitative real-time polymerase chain reaction (RT-PCR) [89]. Integration into present clinical workflows is enabled by compatibility with standard laboratory protocols, thereby reducing the need for specialized equipment. Furthermore, the characterization of circulating miRNAs holds considerable promise for enhancing the management of patients with viral infections, thus providing novel perspectives on disease progression and treatment outcomes. The application of this method could facilitate more precise therapeutic measures and a better understanding of the fundamental processes of viral infection.

MiRNAs are capable of significantly affecting the regulation of immune responses, offering an extra dimension in the organization of immune responses during viral infections [58]. The molecular targets of miRNAs are identified through computational studies. In animals, complementarity between miRNAs and mRNA knockdown is imperfect [90]. The structure of mRNA has also been demonstrated to identify miRNA targets [91,92]. STarMir accounts for mRNA-miRNA binding features based on complex sequence, thermodynamic, and structural characteristics [93].

Numerous studies have indicated that let-7 expression is altered in a variety of viral diseases compared to healthy controls [94]. This work illustrates the involvement of let-7 family miRNAs in regulating multiple mechanisms of immune responses to viral infection, including B-cell responses, and the regulation of both cytokine gene expression and its receptors. Let-7 is involved in the regulation of *PLCG2* (Figure 11A). This gene encodes phospholipase Cg2, which is essential for the function of hematopoietic cells and plays a key role in the regulation of immune responses. Genetic abnormalities affecting this region lead to impaired B-cell differentiation, increased myeloid cell proliferation, and enhanced synthesis of pro-inflammatory cytokines [95,96]. Let-7 is also involved in the regulation of both cytokines and their receptors by interacting with the 3′-UTR region of the corresponding mRNA (Figure 11B,C). According to STarMir data, both let-7a-5p and let-7c-5p bind to transcripts of the *PLCG2*, *IL6*, and its receptor genes at the same site with a high probability of finding the true binding site (LogitProb parameter, Appendix A). let-7c-5p exhibits lower binding energy compared to let-7a-5p for all three transcripts (ΔGhybrid parameter, Appendix A), which may indicate a potentially more stable interaction.

Our data are consistent with literature demonstrating a role for let-7 in viral infections. Let-7 was shown to affect effector CD8+ T cell differentiation and is also associated with B-cell activation and antibody production [97]. Reduced expression levels of let-7a-5p, let-7b-5p, let-7c-5p were reported in COVID-19 patients, spanning both mild and severe cases [98,99]. Additionally, the expression levels of let-7b-5p were observed to be increased in a manner that correlated with the viral load [100]. A significant reduction in let-7e-5p expression was reported in COVID-19 patients requiring oxygenation [101]. The expression of let-7c was observed to increase in the context of ZIKV infection [102].

MiR-146a-5p is involved in the regulation of T-cell response activation during viral infection, regulation of cytokine gene and its receptors, regulation of cytokine signaling pathways. MiR-146a-5p is involved in the regulation of *ICAM1* (Figure 12), which plays a role in T cell activation, migration, and function [103]. This miRNA affects *STAT1*, *IRAK1*, and *TRAF6* (Figure 12). IRAK1 and TRAF6 form a complex in response to signals from the Toll-like receptor/interleukin-1 receptor superfamily (TLR/IL-1R). This leads to the activation of signaling pathways such as NF-κB and MAPK, which in turn can activate *STAT1*, influencing of genes associated with the immune response and inflammation [104,105]. Also, miR-146a-5p has a direct effect on *IL6* (Figure 12). Among these interactions, miR-146a-5p binding to the *TRAF6* transcript is characterized by the highest probability of predicting the true binding site. Moreover, the binding site in the target *IL6* mRNA has the highest structural accessibility (Site_Access parameter, Appendix A).

Earlier studies have demonstrated that numerous viruses can induce the expression of miR-146a-5p, including HIV-1 [106], vesicular stomatitis virus [107], the Epstein–Barr virus [108], and DENV [109]. The molecule miR-146a-5p, also known as inflamma-miR, was shown to regulate inflammation [110,111,112,113,114]. This is corroborated by data on the correlation between miR-146a-5p and IL-6 levels in COVID-19 patients [115]. Clinical data analysis revealed an upregulation of miR-146a-5p during the acute phase of COVID-19 [116], with a notable increase in its expression observed in severe cases [16,117]. DENV infection has also been demonstrated to upregulate miR-146a-5p. This miRNA diminished IFN-β production, thereby promoting the proviral effect of miRNA [109]. HIV-1 infection is associated with increased miR-146a expression in the peripheral blood of infected patients. Higher levels of miRNA have been reported to correlate positively with depletion markers [118]. The induction of miR-146a in enterovirus [119], DENV [109], and Japanese encephalitis virus [120] infected cells has been shown to negatively regulate *TRAF6*, an IFN signaling protein, and consequently, prevents IFN-α/β induction.

MiR-181a-5p is known to be involved in regulating immunoglobulin production, cytokines (IL1A), and the *STAT3* gene-related cytokine signaling pathway (Figure 13). Reduced levels of miR-181a-5p have been demonstrated in cases of severe COVID-19 [121,122]. Additional research indicates that diminished miR-181a-5p levels diminish renin secretion, potentially causing more acute hypotensive episodes in patients [122,123,124]. A decrease in miR-181a expression has been documented in the elderly population [125,126]. It is also noteworthy that the elderly are more vulnerable to viral infections, for example, influenza or COVID-19.

MiR-25-3p, miR-29a-3p, miR-29b-3p, miR-363-3p, and miR-367-3p effect *EOMES* (Figure 14A), which plays a key role in regulating the development and function of T cells, especially CD8+ T cells and memory T cells. EOMES maintains the function of effector T cells by stimulating the expression of cytokines such as IFN-γ and TNF-α [127,128,129,130]. MiR-29b-3p and miR-29b-3p regulate *RAB27A* (Figure 14B). This protein is required for the functioning of natural killer (NK) cells [131,132]. MiR-29b-3p regulates the interleukin-10 receptor beta subunit (IL10RB), which is required for interleukin-10 signaling (Figure 14C). The interaction of miR-367-3p with the *EOMES* transcript is characterized by the highest binding site prediction reliability and structural accessibility values. Furthermore, the binding sites on the *EOMES* and *RAB27* transcripts are the same for miR-29a-3p and miR-29b-3p, and the parameters of these interactions are similar (Appendix A). Serum miR-29a-3p levels have been observed to be inversely proportional to the severity of COVID-19 [133,134]. Decreased miR-29b-3p expression was also observed in lung biopsies from most COVID-19 patients compared to controls [135]. Another study revealed a significant correlation between elevated levels of miR-29a-3p and miR-29b-3p and ZIKV infection [136]. IL-21 produced by CD4+ T cells has been shown to be a significant immunomodulator and promotes miR-29 production during HIV infection through the STAT3 pathway [137,138].

MiR-486-3p is involved in the regulation of *BCL3* (Figure 15A). Bcl-3 is a member of the IκB protein family and an essential modulator of NF-κB activity. It is well established that Bcl-3 is critical for the normal development, survival and differentiation of adaptive immune cells, especially T cells [139]. The predicted binding site of miR-486-3p to the *BCL3* transcript has low confidence (LogitProb < 0.5) and should be interpreted with caution (Appendix A). MiR-486-3p is assumed to modulate T cell activation and differentiation [140], thus corroborating the role of this miRNA in regulating T cell responses demonstrated in this study. The upregulation of miR-486-3p has been demonstrated solely in individuals with severe clinical manifestations of COVID-19 [122,141]. Also, miR-486-5p has been reported to contribute to acute lung injury in COVID-19 [142].

MiR-92a-3p, miR-92b-3p, and miR-98-5p regulate *ICAM1* gene (Figure 15B). MiR-98-5p is involved in regulating of genes important for the immune response, such as *PLCG2*, *IL6*, *IL13*, and *IL10*, and miR-92b-3p-*SOCS5*, *IRAK1*, and *STST3*. The interactions of the studied miRNAs with the *ICAM1* transcript are characterized by similar values of binding site prediction reliability and binding energy. Moreover, the predicted binding site for miR-98-5p has a slightly higher structural accessibility compared to the sites for miR-92a-3p and miR-92b-3p (Appendix A).

Reduced plasma concentrations of miR-92a-3p have been identified in COVID-19 patients with pulmonary fibrosis [143]. A reduction in miR-98-5p expression, prompted by viral elements, has been documented to promote infection initiation [144].

Thus, miRNA profiling permitted the identification of miRNAs implicated in modulating segments of the immune response to viral infection. Presented here is an intricate regulatory network that miRNAs mediate. Variations in the expression of a single miRNA can lead to a cascade of effects that impact various immune system processes.

Integrating our data with existing literature regarding the connection between specific miRNAs and the clinical progression of viral infections enabled us to pinpoint several miRNAs with potential utility as predictive indicators of the clinical course of viral infection. Additionally, spatial omics analysis of various immune cells may illuminate the alterations in gene expression under the influence of miRNAs [145].

### Limitations and Prospects of the Study

The primary limitation of this work is its exclusively computational nature. The results obtained represent valuable predictions rather than experimentally confirmed data. While our analysis identified a number of miRNAs that influence the functioning of the immune system during viral infections. The specific changes in their expression across different diseases and the relationship between the differential expression of these miRNAs and symptom severity remain to be established through further research.

Despite this limitation, the conducted study opens clear avenues for future work. Unlike previous studies, we did not focus on a specific disease, but rather analyzed miRNAs that are involved in regulating the immune response to different types of viral infections. The identified set of miRNAs represents promising candidates for in-depth analysis and experimental validation. The main directions for future research should include:Experimental validation: Validation of the expression of predicted miRNAs using PCR methods [146,147,148], sequencing [149,150], multiplex analysis using optically encoded particles [151], or another method on clinical samples from patients with various viral infections.Clinical Correlation: Analysis of the relationship between the expression levels of the selected miRNAs and clinical parameters, including symptom severity, viral load, and disease outcome.Development of a Diagnostic Panel: Based on the confirmed data—the creation and testing of a diagnostic system or biomarker panel to assess the status of the immune response and predict the course of the disease.

Thus, this work forms an important theoretical foundation and sets a strategic direction for subsequent research. The miRNAs identified in silico are high-probability candidates for the development of new predictive biomarkers and future diagnostic solutions.

## 4. Materials and Methods

### 4.1. Analysis of miRNAs Associated with Viral Infections

The miRWalk database [17] was utilized to identify miRNAs relevant to the pathogenesis of three viral infections: flaviviruses, HIV-1, and coronaviruses (accessed on 18 April 2025). The search was performed using the following identifiers:for flavivirus infections: DOID:0050175#tick-borne encephalitis, DOID:12205#dengue disease, DOID:2366#West Nile fever, DOID:0060478#Zika fever;for coronavirus infections: DOID:0080599#Coronavirus infectious disease, DOID: 0080642#Middle East respiratory syndrome, DOID:2945#Severe acute respiratory syndrome (given the absence of data on other coronavirus infections within the miRWalk database, we deem the analysis to be exhaustive);for HIV infection: DOID:526#human immunodeficiency virus infection disease.

MiRNAs were selected if (1) the predicted probability of functional interaction between the miRNA and the target mRNA was greater than 0.95, (2) the miRNAs bound to the 3′-UTR, and (3) the miRNAs were present in at least one of the databases TargetScan [20], miRDB [21], and miRTarBase [22]. The dataset was processed utilizing the R software version 4.4.2.

### 4.2. Target Predictions of the miRNAs Analyzed

The miRTarBase [31] and TargetScan [32,33] databases were utilized to search for target miRNA genes. This was performed through the CyTargetLinker 4.0.0+ application [34,35] (accessed on 11 April 2025). Visualization of the interaction network was performed using Cytoscape 3.10.2. Gene Ontology (GO) analysis was conducted utilizing the GOlorize 1.0.0 plugin [152]. The gene ontology identifiers (GO_ID) 2312, 2381, 2889, 2286, 2309, 2475, 19883, 19221 and 2323, which are associated with immune system function, were selected. The Benjamini–Hochberg procedure was applied to account for multiple comparisons. False discovery rate (FDR) < 0.05 were deemed statistically significant.

### 4.3. Prediction of miRNA Binding Sites on Target mRNAs

The STarMir web server (https://sfold.wadsworth.org/cgi-bin/starmirWeb.pl accessed on 30 September 2025) was used to predict miRNA binding sites on target mRNAs. The current version of STarMir is an implementation of logistic prediction models developed using high-throughput miRNA binding data obtained by cross-linked immunoprecipitation. The models take into account comprehensive thermodynamics, structure, and sequence [153]. Binding of −15 kcal/mol or lower was considered stable [154].

## 5. Conclusions

This study employed computational approaches to elucidate the significant role of numerous miRNAs in regulating immune responses to viral infections, with several RNA viruses serving as illustrative examples. We examined the regulation of T- and B-cell responses, focusing on processes such as MHC antigen presentation and immunoglobulin production. Detailed consideration was given to the miRNA regulation of cytokines, their receptors, and related signaling pathways. A relationship has already been reported between the severity of a disease and the varying expression levels of the following miRNAs: miR-146a-5p, miR-181a-5p, miR-29a-3p, miR-29a and miR-29b, miR-486-3p, and miR-574-5p. The miRNAs in question are not specific to one infection, but may be associated with a wide range of viral infections. The identified set of miRNAs represents a promising candidate for in-depth analysis and experimental validation as prognostic indicators. Additionally, an understanding of the regulatory pathways of these miRNAs will facilitate the identification of critical connections in the body’s immune system for targeted therapeutic interventions.

Further research should concentrate on defining miRNA expression profiles that are unique to different viruses, infection stages, and the development of complications following infection. The findings will be instrumental in the identification of trustworthy biomarkers for disease diagnosis and observation. A more thorough understanding of the influence of miRNAs on immune responses has the potential to facilitate the development of new therapeutic interventions and proactive strategies in clinical practice.

## Figures and Tables

**Figure 1 ijms-26-10100-f001:**
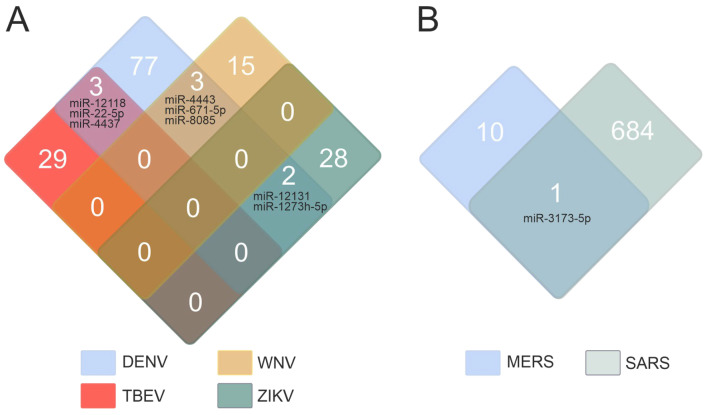
Quantitation of miRNAs associated with flavivirus (**A**) and coronavirus (**B**) infections. The data from the miRWalk database [17] (accessed on 18 April 2025). The figure presents a list of common miRNAs.

**Figure 2 ijms-26-10100-f002:**
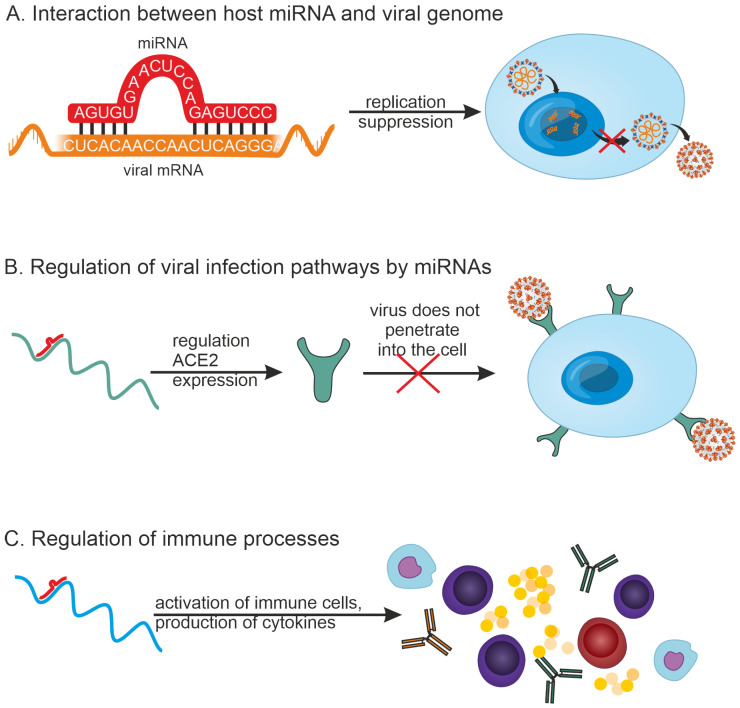
A possible role of host miRNAs during infection. (**A**) MiRNAs can directly bind to the viral RNA genome and inhibit viral replication. (**B**) MiRNAs can be involved in regulating infection pathways, such as regulating the expression of ACE2, which is required for the SARS-CoV-2 virus entry into the host cell. MiRNAs can play a role in regulating multiple components of the immune system. (**C**) MiRNAs play a role in regulating the innate and adaptive immune systems.

**Figure 3 ijms-26-10100-f003:**
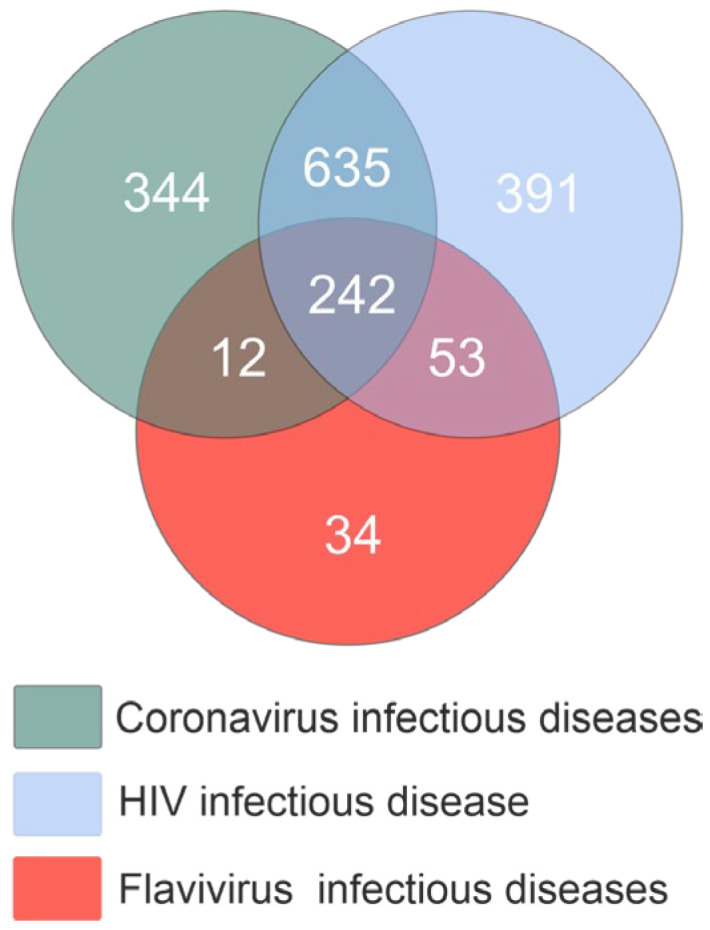
Quantitation of miRNAs associated with three types of viral infections.

**Figure 4 ijms-26-10100-f004:**
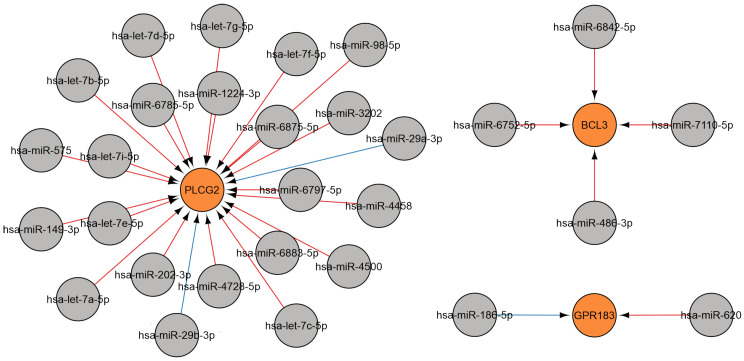
A regulatory network of miRNAs affecting B cell activation involved in the immune response in viral infections. MiRNA data were acquired from miRTarBase [22] (red edges) and TargetScan [20] (blue edges) (accessed on 11 April 2025). Visualization of target genes was performed using Cytoscape 3.10.2. MiRNAs are denoted in gray, and genes are indicated in orange.

**Figure 5 ijms-26-10100-f005:**
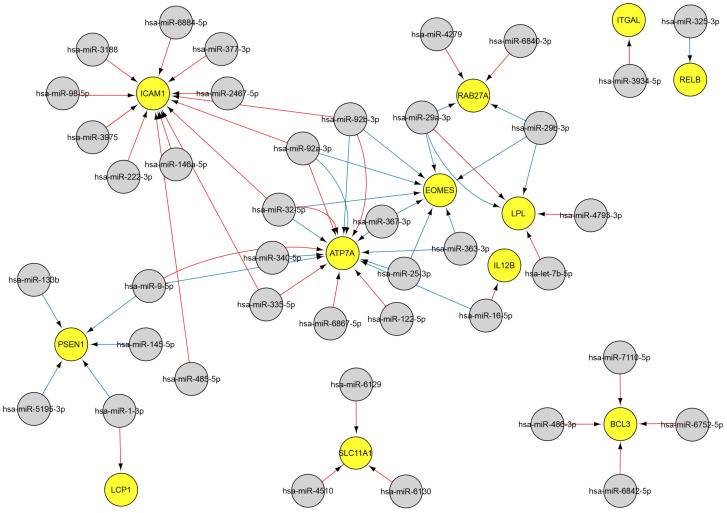
The regulatory network of miRNA-gene interactions during viral infections. MiRNA data were acquired from miRTarBase [22] (red edges) and TargetScan [20] (blue edges) (accessed on 11 April 2025). MiRNAs are indicated by gray, while GO genes are indicated by yellow and blue: 2286 and 2309, T cell activation involved in immune response and T cell proliferation involved in immune response.

**Figure 6 ijms-26-10100-f006:**
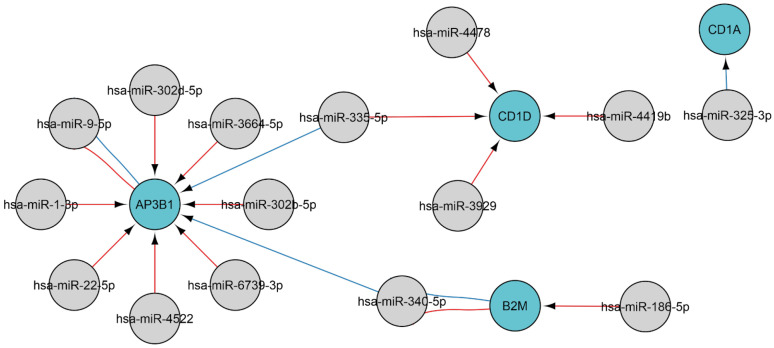
The regulatory network of miRNA-gene interactions in viral infections. MiRNA data were acquired from miRTarBase [22] (red edges) and TargetScan [20] (blue edges) (accessed on 11 April 2025). Selected GO genes: 2475, antigen processing and presentation via MHC. Gray denotes miRNAs, and blue represents genes.

**Figure 7 ijms-26-10100-f007:**
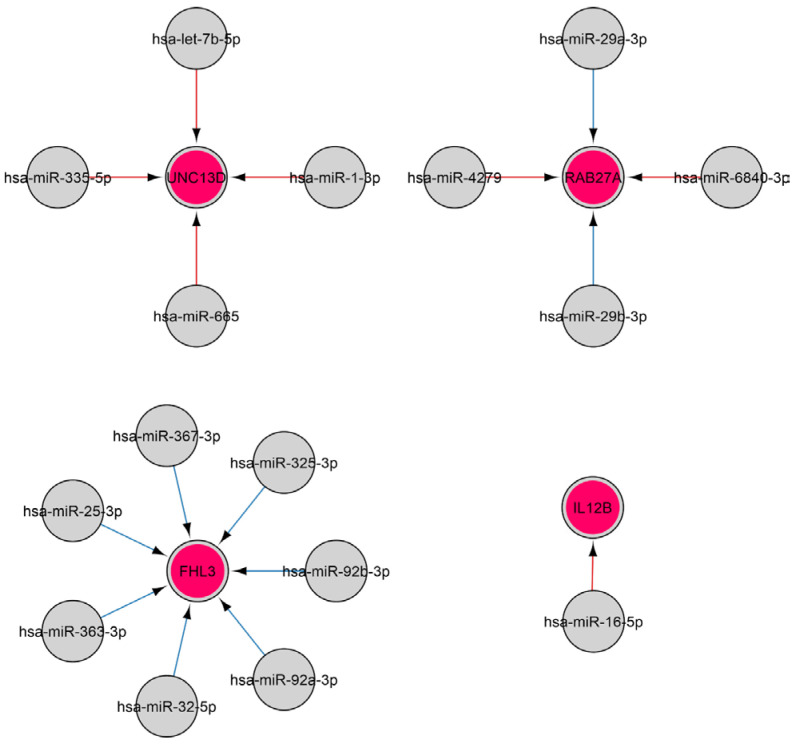
The regulatory network of miRNA-gene interactions in viral infections. MiRNA data were acquired from miRTarBase [22] (red edges) and TargetScan [20] (blue edges) (accessed on 11 April 2025). The selection involved genes associated with natural killer cell activation and immune response (GO: 2323). MiRNAs are denoted in gray, and genes are denoted in red.

**Figure 8 ijms-26-10100-f008:**
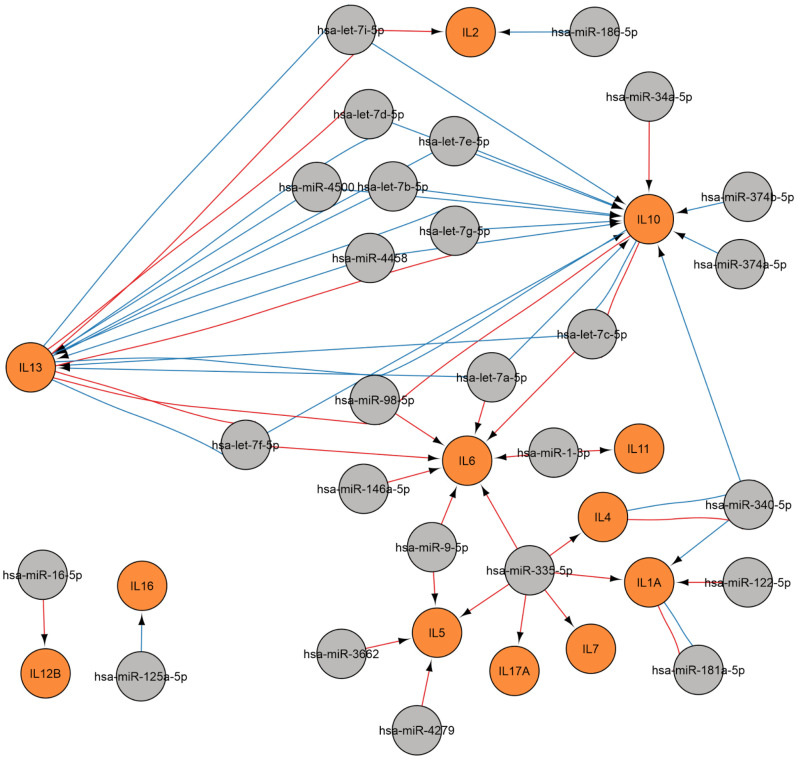
A regulatory network of miRNAs involved in the regulation of cytokine gene during viral infection. MiRNA data were acquired from miRTarBase [22] (red edges) and TargetScan [20] (blue edges) (accessed on 11 April 2025). MiRNAs are indicated in gray, and genes are indicated in orange.

**Figure 9 ijms-26-10100-f009:**
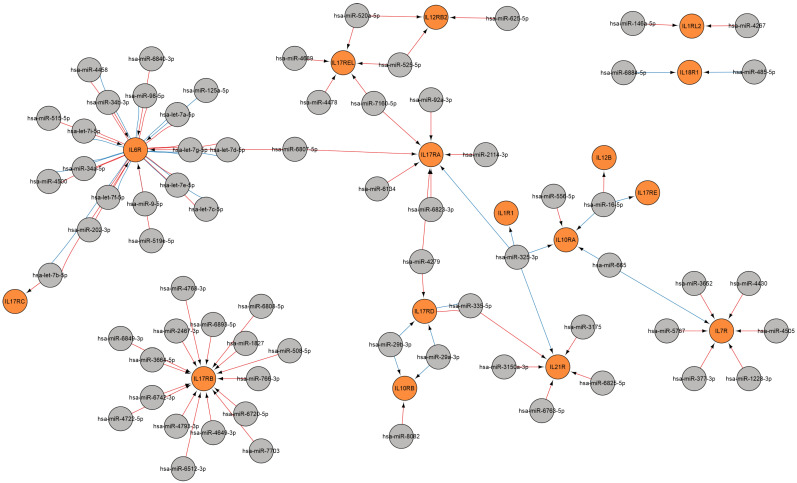
A regulatory network of miRNAs involved in regulating cytokine receptor gene during viral infection. MiRNA data were acquired from miRTarBase [22] (red edges) and TargetScan [20] (blue edges) (accessed on 11 April 2025). MiRNAs are represented in gray, and genes are represented in orange.

**Figure 10 ijms-26-10100-f010:**
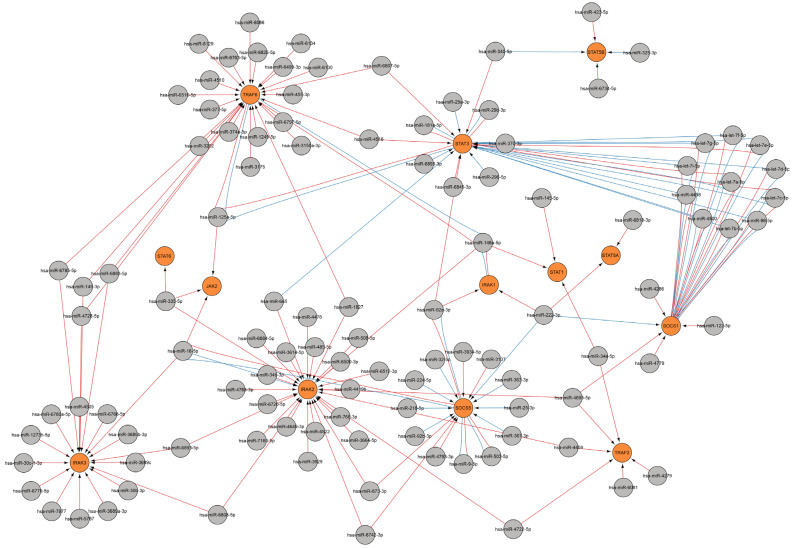
A regulatory network of miRNAs involved in the regulation of cytokine signaling pathways during viral infection. MiRNA data were acquired from miRTarBase [22] (red edges) and TargetScan [20] (blue edges) (accessed on 11 April 2025). MiRNAs are designated in gray, while orange designates genes.

**Figure 11 ijms-26-10100-f011:**
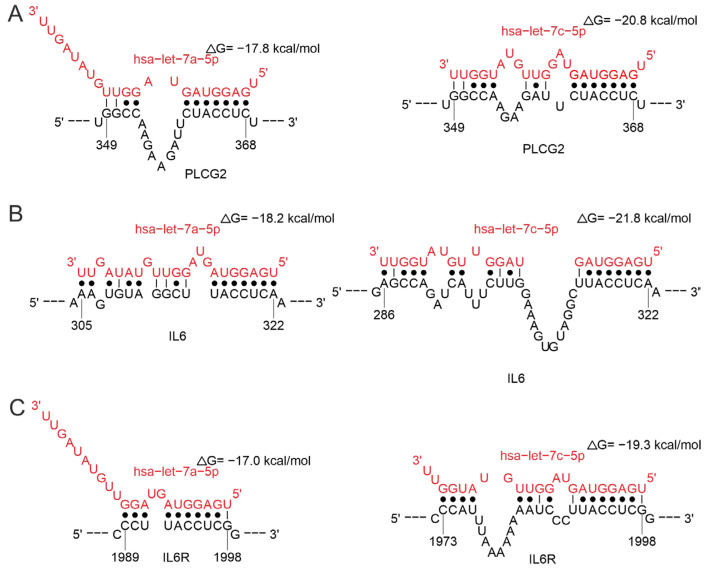
Analysis of let-7 binding in the 3′-UTR region mRNAs of *PLCG2* (**A**), *IL6* (**B**), *IL6R* (**C**) genes.

**Figure 12 ijms-26-10100-f012:**
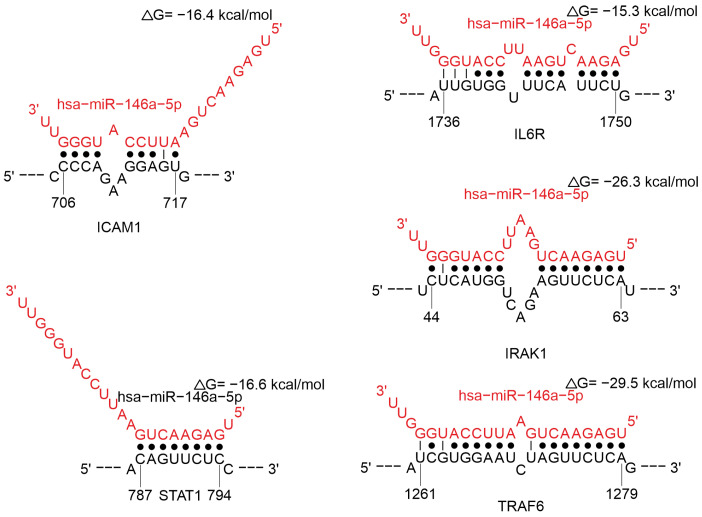
Analysis of miR-146a-5p binding in the 3′-UTR region mRNAs of *ICAM1*, *IL6*, *STAT1*, *IRAK1*, and *TRAF6* gene.

**Figure 13 ijms-26-10100-f013:**
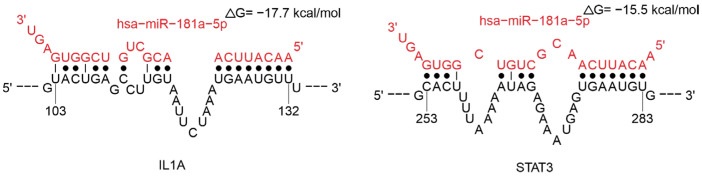
Analysis of miR-181a-5p binding in the 3′-UTR region mRNAs of *IL1A* and *STAT3* gene.

**Figure 14 ijms-26-10100-f014:**
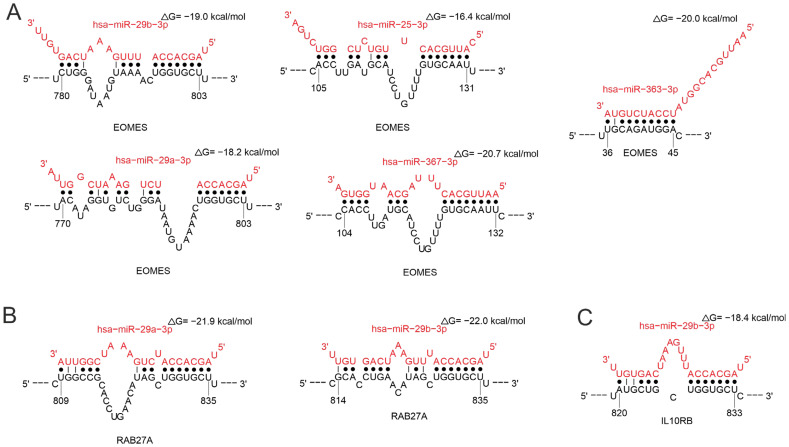
Analysis of miRNA binding in the 3′-UTR region mRNAs of EOMES (**A**), RAB27A (**B**) and IL10RB (**C**) gene.

**Figure 15 ijms-26-10100-f015:**
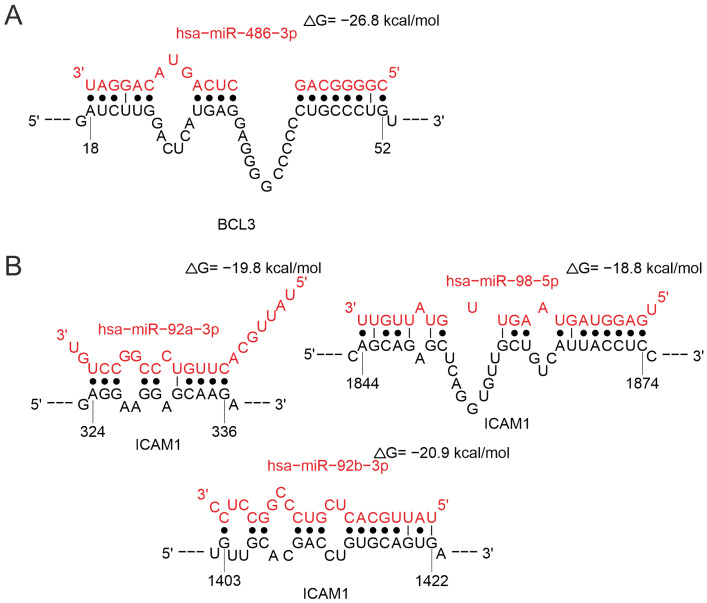
Analysis of miRNA binding in the 3′-UTR region mRNAs of EOMES (**A**) and IL10RB (**B**).

## Data Availability

The original contributions presented in this study are included in the article/Appendix A. Further inquiries can be directed to the corresponding authors.

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
