# Peer review of "In Silico Analysis of MiRNA Regulatory Networks to Identify Potential Biomarkers for the Clinical Course of Viral Infections"

_ijms, 2025, doi:10.3390/ijms262010100_

Round 1

Reviewer 1 Report

Comments and Suggestions for Authors

This work systematically analyzed the immune regulatory network of miRNAs in cross viral infections using bioinformatics methods, focusing on three highly pathogenic viruses of the Flaviviridae family (dengue virus, Zika virus, etc.), Coronavirus (SARS-CoV-2, etc.), and HIV-1. Based on databases such as miRWalk and Targeted Scan, a regulatory network containing 502 nodes (224 miRNAs+278 target genes) and 2236 interactions was constructed. It was clarified that core miRNAs regulate B cell activation, T cell differentiation, NK cell function, and cytokine signaling pathways, affecting the clinical efficacy of viral infections. There are some problems should be resolved before next step, for example:
1. The authors predict that miR-146a-5p targets TRAF6 and miR-181a-5p regulates STAT3 through a database. Additional cell level experiments are needed to clarify the authenticity of the regulatory relationship and avoid false positives caused by relying solely on bioinformatics predictions.
2. It was proposed that miR-29a-3p affects B cell antibody secretion, but this was not validated through in vitro B cell culture experiments (detecting IgM/IgG production after transfection with miR-29a-3p mimetics/inhibitors).
3. Although STarMir was used to predict the binding energy between miRNA and target mRNA, the specific location of the binding site in the 3'UTR (such as whether it is located in the core regulatory region) was not analyzed, and the priority of the same target gene being regulated by multiple miRNAs was not clarified.
4. Only mentioning the differential expression of miRNA in infection, without detecting the expression trend of miRNA through time series experiments, it is impossible to determine its role in early (antiviral immunity) or late (inflammation regulation) infection. 

Author Response

We thank the reviewer for their insightful and constructive comments. We appreciate the time and expertise invested in this review process.

  1. The authors predict that miR-146a-5p targets TRAF6 and miR-181a-5p regulates STAT3 through a database. Additional cell level experiments are needed to clarify the authenticity of the regulatory relationship and avoid false positives caused by relying solely on bioinformatics predictions.

We agree with the comment. This study represents a large-scale in silico analysis aimed at the initial selection of microRNAs that potentially influence immune function during viral infection. We did identify a number of potential regulatory interactions, such as the association of miR-146a-5p with TRAF6 and miR-181a-5p with STAT3, and we consider these to be promising hypotheses for further experimental testing. As noted in the "Limitations and Prospects" section and the title of the paper, this study is based on computational methods, and its main outcome is the formation of a list of priority targets for future experimental work.

  1. It was proposed that miR-29a-3p affects B cell antibody secretion, but this was not validated through in vitro B cell culture experiments (detecting IgM/IgG production after transfection with miR-29a-3p mimetics/inhibitors).

We thank the reviewer for this valuable suggestion. As noted in the response to the previous comment, our work is purely computational. We did not aim to experimentally verify all the predicted interactions, but focused on their systematic analysis and selection of the most likely candidates.

  1. Although STarMir was used to predict the binding energy between miRNA and target mRNA, the specific location of the binding site in the 3'UTR (such as whether it is located in the core regulatory region) was not analyzed, and the priority of the same target gene being regulated by multiple miRNAs was not clarified.

We thank the reviewer for this insightful comment. To further analyze the nature of the predicted interactions, we have added to the Discussion section an analysis of the influence of mRNA sequence, thermodynamics, and secondary structure on the binding efficiency of miRNAs (see lines 419–422, 433–437, 460–462, 501–504, 519–520, 529–533, highlighted in blue). This analysis allows us to more confidently assess the potential efficacy and competition between different miRNAs for binding to a single target.

  1. Only mentioning the differential expression of miRNA in infection, without detecting the expression trend of miRNA through time series experiments, it is impossible to determine its role in early (antiviral immunity) or late (inflammation regulation) infection.

We fully agree with this observation. To avoid misinterpretations and to remain within the scope of our computational approach, we revised the manuscript and removed all references to microRNA expression levels in our study results. Highlighted in blue.

Reviewer 2 Report

Comments and Suggestions for Authors

This manuscript presents a comprehensive computational analysis of miRNA regulatory networks across three major viral infection types. It contributes to a broader understanding of the regulatory roles of miRNAs in viral infections. However, a major revision is required.

Major comments:

  1. The study focuses on miRNA regulatory networks across three viral infections (Flaviviridae, coronaviruses, HIV). While computational integration is valuable, the novelty compared to existing reviews and database-based studies is not fully clear. The authors should explicitly highlight how their approach advances beyond prior works
  2. The study relies exclusively on in silico predictions (miRWalk, TargetScan, miRTarBase). While acceptable for hypothesis generation, the absence of experimental validation is a limitation.
  3. Whenever talking about the interaction between miRNA and mRNA, it would be better to provide a figure to illustrate the base pairing between miRNA and mRNA.
  4. The abstract and conclusions suggest that miRNAs could serve as predictive biomarkers. However, the manuscript does not present any data showing actual expression changes of miRNAs after viral infection..

Minor comments:

  1. Clarify thresholds: Was p < 0.05 corrected for multiple testing in GO analysis? If so, which method (FDR, Bonferroni)?
  2. In some figures, blue edges/lines are shown, but the legend does not clearly state what they represent.

Author Response

Thank you very much for taking the time to review this manuscript. Please find the detailed responses below and the corresponding revisions/corrections highlighted in the resubmitted files.

The authors should explicitly highlight how their approach advances beyond prior works

Unlike previous studies, we did not focus on a specific disease, but rather analyzed microRNAs involved in regulating the immune response to different types of viral infections. These microRNAs are involved in regulating immune processes, and we hypothesize that similar changes in these microRNAs will be observed in other viral infections, including those that humanity may face in the future. The identified set of microRNAs represents promising candidates for in-depth analysis and experimental validation. We have added a "Limitations and Prospects" section and added this information. See lines 539-565.

The study relies exclusively on in silico predictions (miRWalk, TargetScan, miRTarBase). While acceptable for hypothesis generation, the absence of experimental validation is a limitation.

In this study, we conducted a large-scale in silico analysis. This allowed us to select a number of microRNAs that influence immune system function during viral infection. This study identified those microRNAs that are promising for further experimental testing. However, the changes in the expression of these microRNAs in various viral diseases and the relationship between differential expression and symptom severity remain to be determined. We noted this in the "Limitations and Prospects" section.

Whenever talking about the interaction between miRNA and mRNA, it would be better to provide a figure to illustrate the base pairing between miRNA and mRNA. 

We presented figures that illustrate base pairings between miRNA and mRNA. See figures 10-14.

The abstract and conclusions suggest that miRNAs could serve as predictive biomarkers. However, the manuscript does not present any data showing actual expression changes of miRNAs after viral infection. 

In silico predictions identified a number of miRNAs that have potential for further analysis of their differential expression and symptom severity. The data obtained will help identify those microRNAs that are promising as predictive biomarkers and can form the basis for the creation of a diagnostic system. However, these relationships remain to be established. Therefore, we have edited the title, abstract, and conclusions, and added information about future prospects. 

Clarify thresholds: Was p < 0.05 corrected for multiple testing in GO analysis? If so, which method (FDR, Bonferroni)? 

We used FDR correction. We added information, see lines 581-582.

In some figures, blue edges/lines are shown, but the legend does not clearly state what they represent.

We have added the corresponding captions. See Figure 5-9.

Round 2

Reviewer 1 Report

Comments and Suggestions for Authors

The revised manuscript can be accepted in this version.

Reviewer 2 Report

Comments and Suggestions for Authors

I am satisfied by the revised manuscript.